# Measuring Heart Rate Variability Using Facial Video

**DOI:** 10.3390/s22134690

**Published:** 2022-06-21

**Authors:** Gerardo H. Martinez-Delgado, Alfredo J. Correa-Balan, José A. May-Chan, Carlos E. Parra-Elizondo, Luis A. Guzman-Rangel, Antonio Martinez-Torteya

**Affiliations:** 1Programa de Ingeniería Mecatrónica, Universidad de Monterrey, San Pedro Garza García 66238, Mexico; gerardoh.martinez@udem.edu (G.H.M.-D.); alfredo.correa@udem.edu (A.J.C.-B.); jose.may@udem.edu (J.A.M.-C.); carlos.parra@udem.edu (C.E.P.-E.); 2Programa de Maestría en Ingeniería del Producto, Universidad de Monterrey, San Pedro Garza García 66238, Mexico; luis.guzman@udem.edu; 3Escuela de Ingeniería y Tecnologías, Universidad de Monterrey, San Pedro Garza García 66238, Mexico

**Keywords:** Photoplethysmography (PPG), heart rate measurement, Heart Rate Variability (HRV), non-contact, face imaging

## Abstract

Heart Rate Variability (HRV) has become an important risk assessment tool when diagnosing illnesses related to heart health. HRV is typically measured with an electrocardiogram; however, there are multiple studies that use Photoplethysmography (PPG) instead. Measuring HRV with video is beneficial as a non-invasive, hands-free alternative and represents a more accessible approach. We developed a methodology to extract HRV from video based on face detection algorithms and color augmentation. We applied this methodology to 45 samples. Signals obtained from PPG and video recorded an average mean error of less than 1 bpm when measuring the heart rate of all subjects. Furthermore, utilizing PPG and video, we computed 61 variables related to HRV. We compared each of them with three correlation metrics (i.e., Kendall, Pearson, and Spearman), adjusting them for multiple comparisons with the Benjamini–Hochberg method to control the false discovery rate and to retrieve the q-value when considering statistical significance lower than 0.5. Using these methods, we found significant correlations for 38 variables (e.g., Heart Rate, 0.991; Mean NN Interval, 0.990; and NN Interval Count, 0.955) using time-domain, frequency-domain, and non-linear methods.

## 1. Introduction

Heart Rate Variability (HRV), a quantitative marker of autonomic activity that measures the physiological phenomenon of variation in the time interval between heartbeats, has become a very important risk assessment tool. A reduced HRV can be associated with a poorer prognosis for multiple conditions, while more robust periodic changes in the R-R interval show signs of good health [1,2,3,4]. HRV is commonly related to heart health, but it can also provide useful information regarding blood pressure, gas exchange, and gut- and vascular-related matters [5].

According to the American Heart Association, HRV can be measured using time-domain, frequency-domain, and non-linear methods [6], with each method yielding different information for different applications. Time-domain features, such as the mean time between heartbeats (NN interval), the mean heart rate, and the standard deviation of the NN interval, are used to determine the heart rate at any point in time or the intervals between successive heartbeats. Frequency-domain methods are used to obtain information on how variance is distributed as a function of frequency. Some features of interest are the total power (i.e., the variance in NN intervals over the temporal segment) and the power in the very low- (VL), low- (L), and high- (H) frequency ranges (lower than 0.04 Hz, between 0.04 and 0.15 Hz, and between 0.15 and 0.4 Hz, respectively). Lastly, due to the unpredictability of the complex mechanisms that regulate the HRV, non-linear methods are also included because non-linear phenomena are involved.

This study aimed to find a way to obtain HRV measurements from a device other than the main tool currently used, the Holter ECG [7,8,9,10,11]. The alternative that we developed measures blood flow using video to obtain a signal similar to those obtained using Photoplethysmography (PPG), which previous research shows can be used to compute the HRV [12,13,14,15,16,17,18]. With the aid of artificial intelligence (AI), there have been large developments in diagnosis, prognosis, and treatment using visual pattern recognition as a gateway to contribute to the interpretation of images in the medical field [19]. Considering instances when a diagnosis can be prone to human error, such as rare diseases or lack of proper symptoms or omissions, AI and machine learning (ML) can play a big role in reducing the occurrence and probability of a misdiagnosis or overdiagnosis [20]. According to the National Academics of Science, Engineering, and Medicine report from 2015, a majority of people encounter at least one diagnostic mistake throughout their lifespan [21]. The implementation of AI with video, which is explored here, can be developed further with the help of technologies such as the Internet of Things (IoT), which are capable of monitoring patients in real-time and providing timely, effective, and quality healthcare services related to the cardiac condition [22].

The main contribution of this work is that by performing a correlation analysis, we determined that the HRV features extracted from video were comparable in quality to those obtained through PPG when using a database with PPG measurements derived from a pulse oximeter [23]. Significant correlations were found for time- and frequency-domain features as well as for characteristics measured with non-linear methods that index the unpredictability of a time series [24].

## 2. Materials and Methods

To generate our dataset, we used a commercial pulse oximeter to obtain the PPG signal and heart rate of each subject. We also used a camera with a frame rate of 30 fps; this rate permits smooth transitions between frames, which creates a smoother signal output, and a resolution of 1280 × 720 to record the faces of the participants throughout the entirety of the test. The higher the resolution, the more computational cost it presents. The test was designed to have a ten-minute duration in order to capture information in the very-low frequency range. Due to the effect of ballistocardiographic artifacts in PPG signals and how they can affect the obtention of the signal [12], we sat subjects comfortably at a distance between 30 and 50 cm from the camera in a room with ambient light in an attempt to project the least amount of shadows possible to the faces of the subjects. The sample population consisted of 5 subjects: 2 females aged 25 and 55 and 3 males aged 17, 24, and 60, whose skin tones ranged from pearl white to fair to olive. For each subject, we recorded 9 samples at different times of day throughout a 4-month period, amounting to 45 total videos.

Our methodology is shown in Figure 1. We computed HRV features using the PPG data [25] and video independently. For the video signal, we implemented face recognition and color augmentation stages. We also performed time alignment between the PPG and video signals. Once we obtained the features from both signals, we performed a correlation analysis to compute performance metrics.

We used the same methodology to extract HRV features from PPG signals used in our previous work [23], in which HRV was used to determine blood glucose concentration. There, a peak detection algorithm was developed to accurately measure the distance between each peak, and we computed the minimum distance between peaks and their heights to deal with the noise by taking into consideration the low complexity of the signal and its Gaussian behavior (analogous to the R-R interval). Then, the vector of time intervals was used to extract the HRV features using non-linear, time-domain, and frequency-domain methods with the help of the *pyHRV* toolbox [26]. The performance of the peak detection algorithm was evaluated by comparing its output to an annotated PPG database, yielding 99.89% precision. The time-domain methods resulted in 15 features, the frequency-domain methods resulted in 36, and the non-linear methods resulted in 12, yielding a total of 61 of the most commonly measured HRV characteristics [24].

The methodology employed to extract the HRV features from the video began with a face detection stage. Using face landmarks with OpenCV and the *Viola–Jones face detectors variation* from the *Haar cascade classifier* [27], we were able to obtain a region of interest (ROI) from the video (i.e., the face) and cropped all of the frames to focus solely on it. However, in order to minimize the variations between frames, the ROI only changed positions between frames when a large enough translation (experimentally fixed) was detected.

The next stage consisted of a color augmentation method using the Eulerian Video Magnification algorithm developed by Freeman et al. [28] and applied using MATLAB. This stage aimed at amplifying the subtle color variations in the skin as blood fluctuates through it. It decomposes a video sequence into different spatial frequency bands and applies temporal filters to each one, later amplifying them and adding them back to the original signal in order to reveal previously non-visible information.

From the color-augmented video, we only extracted the information from the red channel, given that according to Feng et al. [29], the red channel can capture muscle movement due to the reflection of light on the skin. This signal was then processed in the same way as the PPG signal was, as previously described. Thus, two databases identical in size with information regarding HRV were created, one using PPG data derived from a commercial pulse oximeter, and one using blood-flow-related data derived from video.

Considering that the pulse oximeter study and the video recording did not start and finish at the exact same time, there may have been some misalignment between the two signals. Therefore, we performed a time adjustment, in which the longest signal was trimmed (the same amount at the beginning and at the end of the recording) to match its counterpart. We also stratified the peak detection results on a per-minute basis in order to identify large sources of noise and possible variation between signals.

The mean error (Me), root mean square error (RMSE), and standard deviation (SD) of the heart rate obtained using each database were computed. Additionally, the correlation of each variable between datasets was calculated using the Kendall, Pearson, and Spearman tests. In order to account for multiple comparisons, we used the *Benjamini–Hochberg* method to control the false discovery rate (FDR) by considering statistical significance when the adjusted *p*-value (q-value) was lower than 0.5.

## 3. Results

In this section, we will describe the results obtained at each stage of our methodology since it is crucial to detect any possible errors that could cause reliability issues when measuring HRV, either at the face detection stage, the color augmentation stage, or at the peak detection stage.

### 3.1. Face Detection

Each video was inspected qualitatively when obtaining the ROI. All cropped videos passed our overall quality inspection and showed the forehead, eyes, nose, and part of the mouth of the subjects at almost all times, as shown in Figure 2b. However, some frames did not show a clear image of the face, as shown in Figure 2c, because the subject had turned, either slightly or completely from the camera, or because movements caused blur in the frame, as shown in Figure 2d. Since the video recording had a 10 min duration, it was expected that subjects would turn their face from the camera occasionally.

### 3.2. Color Augmentation

In order to evaluate the performance of the color augmentation stage, we compared the number of peaks detected from the processed video signal to those computed from the PPG signal, both globally and locally (1 min windows). However, we first performed a qualitative inspection to make sure that the red channel of the processed video yielded the expected results (i.e., an oscillating signal with a frequency within normal heart rate values). Additionally, we visually compared the shape and period of the video and PPG signals, as shown in Figure 3.

### 3.3. Peak Detection

After retrieving the signals from both the pulse oximeter and the video, we located the peaks of each sample. We also trimmed the longest signal of each pair of results per observation in order to align them time-wise. Figure 4 shows the results for one observation where it can be seen that the peak detection algorithm was accurate. Using the number of peaks, the sample rate of the pulse oximeter, and the fps of the video signal, we calculated the heart rate for each sample in beats per minute (bpm).

Table 1 shows that there was less than 1 bpm of difference on average between the heart rate measured from the pulse oximeter and the one measured from the video. In terms of total peaks, there was an average difference of 12.804 peaks per 10 min between signals.

Additionally, we performed the same comparison per 1 min window in order to detect specific time lapses in which there were significant differences between both signals. Table 2 shows that the biggest differences are in the first two minutes, with more than 2 bpm of difference between both means of measurement and a standard deviation in the first minute of more than 3 bpm. In addition, the table shows that at the last minute, there is an increase in the difference of more than 1.8 bpm.

### 3.4. Correlation Analysis

After computing the overall performance metrics, we obtained the correlation of each variable between both datasets. In total, 38 features were regarded as having a significant correlation between sources under the three different tests after adjusting for multiple comparisons. Table 3 shows the five features with the highest correlations under the Pearson test, the most commonly used metric for general correlation between variables of this kind [30]; it also includes the correlation coefficients (r), q-values, and the method used to measure it.

Furthermore, Table 4 represents the five features with the lowest q-values, which also share the trait of having the highest correlation under the Kendall test; it also includes the method used, correlation coefficients (r), and q-values.

Finally, Table 5 shows the five features with the lowest q-values and highest correlation under the Spearman test, presenting the same five features as in the previous tests; it also includes the corresponding method used to measure it, correlation coefficients (r), and q-values.

For the time-domain methods, the nine features that were significant under the three tests after adjusting for multiple comparisons can be seen in Table 6.

For the frequency-domain methods, there were 26 significant features. Table 7 shows the significant features retrieved when using the *Welch* method to calculate the power spectral density (PSD).

Table 8 shows the significant features obtained when using the *autoregressive* method to calculate the PSD.

Regarding the use of the *Lomg–Scargle* method to calculate the PSD, Table 9 presents the significant features obtained from this method.

Lastly, there were three significant features from the non-linear methods, all derived from a Poincaré plot, as seen in Table 10.

## 4. Discussion

Our results show that the signal derived from the video had a very similar composition in terms of the locations of peaks to the signal derived from the commercial pulse oximeter, as determined by their smaller than 1 bpm average difference. To truly test the performance of the peak detection method we proposed, a comparison was made to other previously developed methods. Table 11 shows the comparison between our proposed method and other methods, such as those involving deep learning [31] and neural networks [32], with the commonly used metrics to evaluate their performance [33].

From Table 11, we can see that our proposed methodology is first in both RMSE and Me and ranked fourth in SD, which leads us to believe that the information and setup we are using can be reliable when it comes to computing HRV features.

After performing a deeper analysis, in which the results were compared in 1 min windows, there was a tendency for the video to compute more peaks at the start and end of each study. This is probably due to the subjects settling into their chairs at the beginning of the study or preparing to get up by the end of it. These situations can lead to the video being cropped poorly, therefore obtaining a signal that does not relate to the skin of the subject, causing the signal from the R channel to provide false information. Nonetheless, even when taking this situation into account, the computation of heart rate was accurate, which means that when taking the correct time intervals and after cleaning the signal, we were able to compute the heart rate of a person as well as a pulse oximeter can.

Additionally, we demonstrated that we were able to measure 38 HRV features with a significant correlation between sources of information; we considered three correlation tests and *p*-value adjustment for multiple comparisons. The features extracted from the PPG signal derived from the pulse oximeter were significantly correlated to those derived from the video. Moreover, those features included information generated using multiple methods: time-domain methods, frequency-domain methods, and non-linear methods. Specifically, in the frequency-domain, there were a considerable number of useful correlations in the variables related to Welch’s periodogram, a method used for spectrum monitoring [36,37], and improvements when measuring HRV with alternative methods [38,39]. In addition, there was a considerable number of parameters related to autoregressive models that mainly focus on spectral analysis [39] and that can also be related to other non-parametric analyses [40]. For the non-linear metrics, there were also significant correlations in the Poincaré plot for both axes and their ratios. These metrics bring valuable information to variability in R-R Intervals and other measures of variability [41,42,43] and have been previously related to PPG as well [44].

This work has some limitations; we only used information from one of the three channels of the video, while some authors have recommended mixing both the red and green channels to filter the noise of sudden movements from the subject due to the wavelength dependency of reflection PPG and optics [29]. Another cause of error could be related to the ROI selected in this study, as the whole face of the subject was taken into account and there could have been a smaller ROI focused on the forehead or another large area of the face to prevent losing information when cropping the frame. Finally, the population size is small, and thus, these results are yet to be generalizable; we intend to increase the population size and validate these results. Another area of opportunity is the assessment of natural light quality to eliminate the noise resulting from changes in light. Authors recommend using more robust models during imaging [45,46] that use deep learning methods and statistical approaches for natural spatiotemporal scenes, which would help with frame cropping and improve the quality of the exploited video.

## 5. Conclusions

In this work, we demonstrated that the HRV information derived from a PPG signal is significantly correlated to those same features when derived from a video. We performed 45 studies that included a 10 min session in which the subjects were connected to a pulse oximeter and looked into a camera. Peaks were detected from the PPG signal that the pulse oximeter yielded and were validated against a manually annotated dataset. Peaks were detected from the R channel of the video using the same methodology after a face detection and color augmentation stage that aimed at focusing on the face of the subjects and amplifying the subtle color changes in the skin caused by blood flow. On average, an error of less than 1 bpm was found when calculating the heart rate using each signal independently. Furthermore, HRV features were extracted from each signal using time-domain, frequency-domain, and non-linear methods. When compared using Pearson, Spearman, and Kendall correlations tests, and after adjusting for multiple comparisons, 38 of the 61 extracted features had a significant correlation in all three tests, including features from the three types of methods tested. These results are promising when it comes to possible applications, as this methodology could be employed in experiments related to the regulation of autonomic balance, blood pressure, gas exchange, and respiratory rate, as well as gut-, heart-, and vascular-related issues.

## Figures and Tables

**Figure 1 sensors-22-04690-f001:**
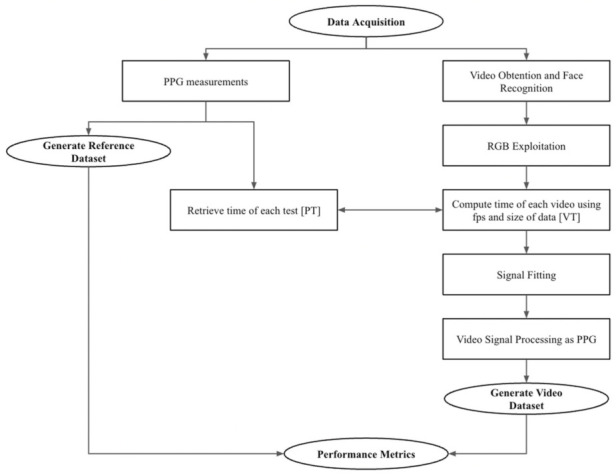
Flowchart depicting pulse oximeter and video processing to generate the datasets that will be used to compute performance metrics.

**Figure 2 sensors-22-04690-f002:**
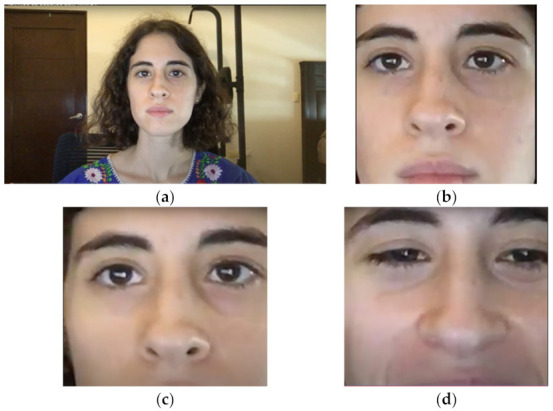
(**a**) Frame from original video taken from camera; (**b**) cropped frame according to ROI; (**c**) image cropped left side of face; (**d**) blurred frame caused by sudden movements.

**Figure 3 sensors-22-04690-f003:**
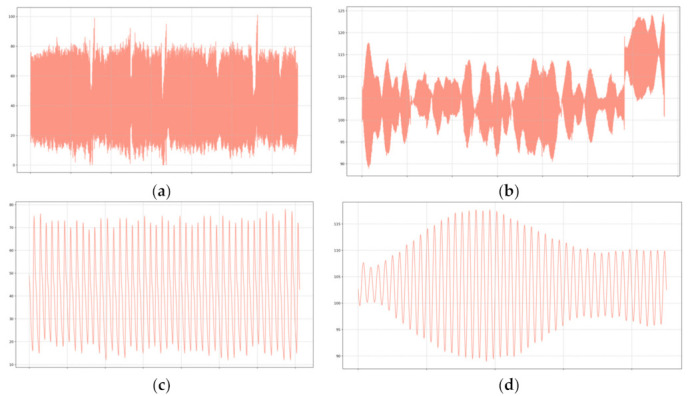
(**a**) Raw PPG signal; (**b**) red channel of the color-augmented video signal; (**c**) half-minute interval of the raw PPG signal; (**d**) half-minute interval of the color-augmented video signal.

**Figure 4 sensors-22-04690-f004:**
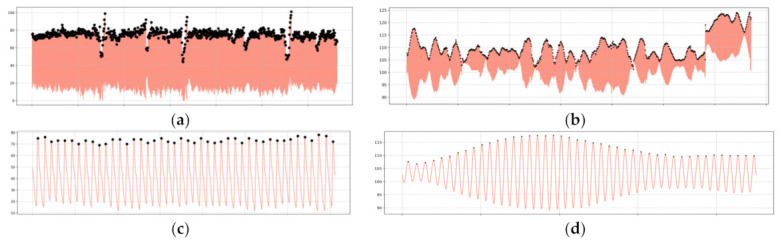
(**a**) Raw PPG signal with peak detection; (**b**) color-augmented video with peak detection; (**c**) half-minute interval of the raw PPG signal with peak detection; (**d**) half-minute interval of the color-augmented video with peak detection.

**Table 1 sensors-22-04690-t001:** Metric computation for heart rate.

Me (bpm)	SD (bpm)	RMSE (bpm)
0.8084	7.7758	1.0971

**Table 2 sensors-22-04690-t002:** Metric computation for heart rate minute by minute.

Minute Interval	Me in bpm (SD)
0:00–0:59	2.5308 (3.5017)
1:00–1:59	2.0444 (2.2049)
2:00–2:59	1.7841 (1.7617)
3:00–3:59	1.9370 (2.1008)
4:00–4:59	1.6222 (1.4815)
5:00–5:59	1.6500 (1.4966)
6:00–6:59	1.6667 (1.7965)
7:00–7:59	1.5333 (1.8165)
8:00–8:59	1.8667 (1.5609)
9:00–9:59	1.9333 (2.0158)

**Table 3 sensors-22-04690-t003:** Top five features according to the Pearson correlation test.

Feature	Method	r	q-Value
Heart Rate	Time-domain	0.991	2.71 × 10^−34^
Mean NN Interval	Time-domain	0.990	1.85 × 10^−33^
NN Interval Count	Time-domain	0.955	4.44 × 10^−21^
Logarithmic VL Frequency Power	Frequency-domain (autoregressive)	0.653	3.57 × 10^−5^
Absolute VL Frequency Power	Frequency-domain (autoregressive)	0.652	3.57 × 10^−5^

**Table 4 sensors-22-04690-t004:** Top five features according to the Kendall correlation test.

Feature	Method	r	q-Value
Heart Rate	Time-domain	0.934	5.09 × 10^−16^
Mean NN Interval	Time-domain	0.919	8.18 × 10^−16^
NN Interval Count	Time-domain	0.879	1.44 × 10^−14^
Logarithmic VL Frequency Power	Frequency-domain (autoregressive)	0.507	3.93 × 10^−5^
Absolute VL Frequency Power	Frequency-domain (autoregressive)	0.507	3.93 × 10^−5^

**Table 5 sensors-22-04690-t005:** Top five features according to the Spearman correlation test.

Feature	Method	r	q-Value
Heart Rate	Time-domain	0.990	8.27 × 10^−34^
Mean NN Interval	Time-domain	0.987	2.55 × 10^−31^
NN Interval Count	Time-domain	0.962	1.43 × 10^−22^
Logarithmic VL Frequency Power	Frequency-domain (autoregressive)	0.624	1.21 × 10^−4^
Absolute VL Frequency Power	Frequency-domain (autoregressive)	0.624	1.21 × 10^−4^

**Table 6 sensors-22-04690-t006:** Significant features regarding time-domain methods.

# of Feature	Feature
1	Heart Rate
2	Root Mean Square of Successive NN Interval Differences
3	SD of NN intervals
4	Percentage of Successive NN Intervals that differ by more than 20 ms
5	Successive NN Intervals that differ by more than 50 ms
6	NN interval count
7	Minimum NN interval
8	Mean NN interval
9	Mean Difference of Successive NN intervals

**Table 7 sensors-22-04690-t007:** Significant features regarding frequency-domain using the Welch method.

# of Feature	Feature
1	Peak VL Frequency Power
2	Absolute VL Frequency Power
3	Relative VL Frequency Power
4	Logarithmic VL Frequency Power
5	Absolute L Frequency Power
6	Logarithmic L Frequency Power
7	Logarithmic H Frequency Power

**Table 8 sensors-22-04690-t008:** Significant features in frequency-domain using the autoregressive method.

# of Feature	Feature
1	Absolute VL Frequency Power
2	Relative VL Frequency Power
3	Logarithmic VL Frequency Power
4	Logarithmic L Frequency Power
5	Absolute L Frequency Power
6	Absolute H Frequency Power
7	Relative H Frequency Power
8	Logarithmic H Frequency Power

**Table 9 sensors-22-04690-t009:** Significant features in frequency-domain using the *Lomg–Scargle* method.

# of Feature	Feature
1	Peak VL Frequency Power
2	Absolute VL Frequency Power
3	Relative VL Frequency Power
4	Logarithmic L Frequency Power
5	Absolute L Frequency Power
6	Absolute H Frequency Power
7	Relative H Frequency Power
8	Logarithmic H Frequency Power
9	Peak H Frequency Power

**Table 10 sensors-22-04690-t010:** Significant features from non-linear methods.

# of Feature	Feature
1	SD perpendicular to the line of identity (SD1)
2	SD along the line of identity (SD2)
3	SD1 to SD2 ratio

**Table 11 sensors-22-04690-t011:** Results of comparison with previous methods.

Citation	Me in bpm (SD)	RMSE (bpm)
Li et al., 2014 [34]	7.14 (9.53)	12.47
Lam et al., 2015 [35]	6.49 (8.54)	10.34
Feng et al., 2015 [29]	6.64 (8.01)	10.12
Haque et al., 2016 [18]	4.69 (3.43)	5.96
Song et al., 2020 [32]	5.98 (7.31)	7.45
Hsu et al., 2020 [31]	−2.07 (4.23)	3.08
Proposed	0.81 (7.77)	1.10

## Data Availability

The data presented in this study are available on request from the corresponding author.

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
