# Peer review of "Measuring Heart Rate Variability Using Facial Video"

_sensors, 2022, doi:10.3390/s22134690_

Round 1
Reviewer 1 Report
The paper is well written, and I believe with small modifications and some elaboration this paper will draw attention toward a large audience.
Here are the few minor suggestions that I want authors to address:
Can you provide some statistical details in the abstract?
While the author suggested some good approach which contribute in image processing domains as well, I would suggest the author should cite the following articles in order make the introduction part a little bit longer.
1) Ahsan, M. M., Luna, S. A., & Siddique, Z. (2022, March). Machine-Learning-Based Disease Diagnosis: A Comprehensive Review. In Healthcare (Vol. 10, No. 3, p. 541). MDPI.
2) Tran, B. X., Latkin, C. A., Vu, G. T., Nguyen, H. L. T., Nghiem, S., Tan, M. X., ... & Ho, R. (2019). The current research landscape of the application of artificial intelligence in managing cerebrovascular and heart diseases: A bibliometric and content analysis. International journal of environmental research and public health, 16(15), 2699.
3) Umer, M., Sadiq, S., Karamti, H., Karamti, W., Majeed, R., & Nappi, M. (2022). IoT Based Smart Monitoring of Patients’ with Acute Heart Failure. Sensors, 22(7), 2431.
Can you please take care of Table 3? Please change those q value into numeric value (i.e., 0.003, 0.0002). Cause for a general audience it will hard to follow if they look into the scientific value.
Best of luck with the papers!
Reviewer 2 Report
Based on facial video, this paper investigates the measure of heart rate variability using PPG. Some comments are listed as follows:
1. The main contributions are suggested to be summarized at the end of the introduction part.
2. As we know that the signals from PPG could be noisy. Please clarify how do you deal with the acquired data.
3. The selection method of features is missing, which could be vital for the paper. And how do you combine these features for the final results.
4. Does the video quality affect the results? Except for the video heat rate measurement, natural video quality assessment is suggested to be reviewed, including No-reference video quality assessment using natural spatiotemporal scene statistics, Deep local and global spatiotemporal feature aggregation for blind video quality assessment, etc.
5. It would be better to provide a table to summarize the used features. Some figures can be presented to demonstrate the relationship between features and heart rate.
6. The experimental results are confusing. For example, in Table 2, where is the correlation? What correlation is used in Table 3? It is recommended to include all three correlations.
7. To validate the performance of the proposed model, state-of-the-art methods could be compared in the experiments.
Round 2
Reviewer 2 Report
The authors have addressed my comments.